# Towards Resource-Efficient LLMs: End-to-End Energy Accounting of Distillation Pipelines

**Katherine Lambert** [1]  **Sasha Luccioni** [2]

## Abstract

The rise in deployment of large language models has driven a surge in GPU demand and datacenter scaling, raising concerns about electricity use, grid stress, and the impacts of modern AI workloads. Distillation is often promoted as one of the most effective paths to obtain cheaper, more efficient models, yet these claims rarely account for the full end-to-end energy and resource costs, including crucial teacher-side workloads such as data generation, logit caching, and evaluation. We present a comprehensive energy accounting framework that measures the complete computational cost of distillation pipelines via detailed stage-wise tracking of GPU device power consumption. In our experiments, we separate and log empirical energy use across distinct phases and systematically measure the energy and emissions of two common distillation methods: the classic logit-based knowledge distillation and synthetic-data supervised fine-tuning, constructing energy–quality Pareto frontiers that expose the previously ignored costs. From these measurements and analyses, we derive practical design rules for selecting distillation methods and hyperparameters under energy and budget constraints, and release an open-source measurement harness and accounting protocol to provide a standardized foundation for comparable, reproducible distillation research, explicitly accountable for complete pipeline energy impact.

## 1. Introduction

The rapid deployment of large language models (LLMs) has accelerated GPU demand and datacenter expansion, intensifying concerns about electricity use, grid stress, and environmental impact. From a "Green AI" perspective, energy should be evaluated alongside accuracy and performance. Distillation is often framed as a primary lever for reducing compute by producing smaller students that preserve much of a teacher's quality while enabling cheaper inference. However, it can require substantial teacher-side work—synthetic data generation, logit caching, filtering, and evaluation—that can rival or exceed student training, especially under hyperparameter sweeps (e.g., KD temperature, decoding settings, filtering thresholds). Prior work typically reports student-side savings (FLOPs, runtime, or training energy) while omitting these upstream costs, making sustainability and cost claims hard to substantiate.

We address this gap with an end-to-end, distillation-specific energy accounting framework that (i) delineates pipeline stages, (ii) measures energy, quality, and throughput per stage under consistent protocols, and (iii) incorporates teacher-side costs in comparisons across pipelines. Our analysis is organized around three questions:

1. When do distillation pipelines (KD or synthetic supervised fine-tuning (SFT)) deliver better energy–quality tradeoffs than a strong SFT baseline under fixed hardware and training budgets?

2. How do teacher-side costs (generation, logit caching, evaluation) compare to student training, and when do they dominate the overall energy budget?

3. Under what conditions — student scale, sequence length, teacher reuse, and target quality — does distillation actually reduce end-to-end energy use and emissions relative to alternatives?

To answer these questions, we make three contributions:

- **Distillation energy protocol and harness.** We formalize distillation stages, logging rules, and metrics to produce comparable energy charts for KD and synthetic SFT, using NVML-based GPU energy as ground truth and empirical CPU/$CO_2$e estimates.

- **Controlled benchmark with stage-wise accounting.** Under fixed hardware, software, and training budgets, we

[1]University of Toronto [2]Sustainable AI Group. Correspondence to: Katherine Lambert <katherine.lambert@mail.utoronto.ca>.

*Proceedings of the 43rd International Conference on Machine Learning*, Seoul, South Korea. PMLR 306, 2026. Copyright 2026 by the author(s).

benchmark 1B/7B/13B OLMo-2 students and report stage-wise energy, runtime, and J/token, constructing Pareto frontiers that separate regimes in the energy–quality space.

- **Design rules and break-even conditions.** We quantify how teacher reuse, sequence length, and key hyperparameters (KD temperature/loss weight; decoding and synthetic-data reuse) shift these frontiers, and derive conditions where distillation is truly cheaper in energy/emissions—and where it is not.

Taken together, our results provide a distillation-specific lens on economical and sustainable AI: rather than treating smaller students as automatically efficient, we show how to account for the full pipeline, measure its energy costs, and decide when distillation is an energy- and resource-efficient choice.

## 2. Background and Related Work

The energy and carbon costs of modern ML systems have motivated the "Green AI" perspective, which argues that efficiency should be evaluated alongside predictive performance (Schwartz et al., 2020; Strubell et al., 2019; Patterson et al., 2022). Prior work also emphasizes the need for transparent reporting of hardware, runtime, and location assumptions, and for careful treatment of measurement uncertainty (Henderson et al., 2020). Our protocol follows these recommendations, but focuses on *directly measured* energy for distillation pipelines (Section 4) rather than proxy metrics such as GPU-hours or FLOPs.

**Energy accounting for distillation.** Recent life-cycle reviews of AI environmental reporting identify post-training adaptation as a major blind spot. Despite the growing use of fine-tuning, distillation, quantization, and related methods, these methods are rarely measured separately, and the energy invested in creating distilled or compressed models is rarely reported or weighed against downstream inference savings (Lambert & Luccioni, 2026).

A small body of work examines the environmental cost and impacts of distillation. Rafat et al. show that KD for CNNs can be carbon-intensive and highlight the importance of energy-aware tuning (Rafat et al., 2023). Yuan et al. compare distilled and non-distilled NLP models, focusing on inference-time energy and runtime while largely treating teacher training and the distillation pipeline as sunk costs (Yuan et al., 2024). These results suggest distillation is not inherently "green," but existing studies typically do not provide a stage-wise, end-to-end holistic accounting that separates teacher-side workloads (e.g., generation/logit caching, filtering) from student training and evaluation under a consistent budget, and provides suggestions on how to

reduce it.

**Measurement tools and methodology.** Energy reporting commonly relies on estimation toolchains such as Code-Carbon and Experiment Impact Tracker (Courty et al., 2024; Henderson et al., 2020), alongside work comparing telemetry-based measurements with model-based estimates and documenting estimator error modes (Bouza et al., 2023; Bannour et al., 2021). These tools motivate a fidelity–convenience trade-off: estimator-only approaches are easy to deploy but can misestimate device energy, while telemetry-based logging requires tighter integration. Our framework combines NVML-based GPU telemetry as the ground-truth signal with lightweight estimators for CPU energy and $CO_2$e, packaged into a reusable harness with explicit stage boundaries and logging rules.

## 3. Distillation Pipeline Setup

We study three standard regimes: baseline supervised fine-tuning (SFT), logit-based knowledge distillation (KD), and synthetic supervised fine-tuning (synthetic SFT) on a single, fully open LLM family. All runs share the same hardware, software stack, tokenizer, and data preprocessing.

### 3.1. Hardware and Environment

All of our experiments were run on a single NVIDIA H100 SXM 80 GB GPU and 16 Intel Xeon Gold 6442Y CPU cores, on exclusive nodes to avoid noise from jobs running on neighboring GPUs. The software environment (Linux distribution, kernel, NVIDIA driver, and CUDA/package versions) was fixed across all runs. Logging is versioned by a Git commit record for each experiment. A full configuration table appears in the Appendix; measurement tools and logging details are described in Section 4, controlling the environmental drift so that observed energy differences arise purely from pipeline structure, model size and hyperparameter differences.

### 3.2. Models and Tasks

Table 1 summarizes the models used in teacher/student distillation. We chose the OLMo-2 family for its fully open weights and training methodology, a shared tokenizer and pretraining lineage across sizes, and existing environmental and energy accounting, allowing us to represent the full lifecycle of energy use of the model. All models share the same tokenizer, and we tokenize prompts and outputs identically for teacher and students, enabling comparable token-count and Joules-per-token statistics.

To avoid domain-specific conclusions, we instantiate pipelines on three supervised workloads: instruction following (TULU-3 SFT mixture) (Lambert et al., 2024), math rea-

*Table 1.* Teacher and student models used in our experiments.

| Role | Model (Hugging Face ID) | Params |
|------|-------------------------|--------|
| Teacher | `allenai/OLMo-2-0325-32B-SFT` | 32B |
| Student | `allenai/OLMo-2-0425-1B` | 1B |
| Student | `allenai/OLMo-2-1124-7B` | 7B |
| Student | `allenai/OLMo-2-1124-13B` | 13B |

soning (OpenR1-Math-220k) (Zhao et al., 2025), and code generation (Open-R1 Codeforces) (Penedo et al., 2025). All datasets are formatted through a single OLMo-2–compatible chat-style prompting pipeline. TULU serves as our primary setting for most experiments, while the math and code workloads act as robustness checks. Across otherwise identical runs, we observed only minor differences in end-to-end energy between datasets relative to the much larger effects of pipeline choice and student scale. In Section 6, we therefore report the main energy metrics (kWh and J/token) as averages over datasets for each pipeline–size configuration.

### 3.3. Distillation Pipelines

We consider two regimes that mirror the most common distillation methods used in practice for deployment-oriented LLMs: logit-based KD and synthetic SFT. Across all regimes, optimizer, schedule, precision, effective batch size, and early stopping are held fixed (see Section 5). The three pipelines below — direct SFT, logit-based KD, and synthetic sequence distillation – provide a minimal but representative set of distillation patterns under a shared, controlled environment, supporting the stage-wise energy frontiers and break-even analyses in the following sections.

#### 3.3.1. LOGIT-BASED KNOWLEDGE DISTILLATION (KD)

This represents the traditional KD regime as described by Hinton et al. (Hinton et al., 2015), where the student is trained to match the teacher's token-level distribution via offline distillation in two stages:

**Teacher logit caching.** For a fixed training corpus, the teacher (a 32B model) is run in inference mode, generating samples from the inputs in the given dataset. For each token position we cache the top-$k$ logits ($k = 100$) and indices.

**Student KD training.** Given the cached teacher distributions $p_t$ and hard labels $y_{\mathrm{hard}}$, the student distribution $p_s$ is trained with:

$$\mathcal{L}_{\mathrm{KD}}(\theta_s) = \alpha\,\mathrm{CE}\big(y_{\mathrm{hard}}, p_s\big) + (1-\alpha)\,T^2\,\mathrm{KL}\big(p_t^{(T)} \,\|\, p_s^{(T)}\big), \quad (1)$$

where $T$ is the distillation temperature, $p^{(T)}$ denotes distributions softened by $T$, and $\alpha \in [0, 1]$ trades off hard-label supervision and soft-label matching. Our core grid uses a default choice of $\alpha = 0.5$ and $T = 1$; both are varied in

sensitivity experiments (see Section 6).

#### 3.3.2. SYNTHETIC SUPERVISED FINE-TUNING (SFT)

Often called the "cheaper" distillation method, synthetic SFT implements data or sequence distillation, i.e. the teacher generates outputs that are then treated as hard labels for supervised fine-tuning of the student.

**Teacher data generation.** For each dataset, we keep the original prompts (instructions, math problems, programming tasks) and replace original dataset labels with teacher-generated responses, using a standardized decoding configuration (nucleus sampling with fixed top-$p$, temperature, and maximum length; see Section 5). This generation pass is run once per dataset, and the resulting synthetic corpora are reused across student sizes and hyperparameter settings.

**Student synthetic SFT.** Students are fine-tuned on the synthetic datasets using the standard autoregressive cross-entropy objective:

$$\mathcal{L}_{\mathrm{SFT}}(\theta_s; x, y) = -\sum_{t=1}^{s} \log p_{\theta_s}\big(y_t \mid x, y_{<t}\big), \quad (2)$$

where $x$ is the input prompt, $y = (y_1, \ldots, y_s)$ is the target sequence, $\theta_s$ are the student parameters, and $p_{\theta_s}$ is the student conditional token distribution. In the synthetic SFT regime, the targets $y_t$ are teacher-generated continuations of the original prompts.

#### 3.3.3. BASELINE SUPERVISED FINE-TUNING

To provide a non-distilled reference, we also train the same students in a baseline SFT regime, where the model is trained directly on the original dataset labels using the same objective as in Eq. 2, but with original dataset target labels $y$. We use a single global training budget and early stopping based on validation loss, and select the best checkpoint for evaluation. This baseline anchors both quality and energy, and serves as a control for measuring the incremental cost and potential gains of introducing a teacher.

## 4. Energy Accounting

We adopt a distillation-specific energy accounting protocol aligned with best-practice recommendations for systematic reporting of energy in machine learning (Henderson et al., 2020), tailored to the design of our pipelines. The goal is to measure *end-to-end* energy in a way that is reproducible, reveals the energy load distributions across different stages of distillation pipelines and is comparable across model sizes and experiments.

### 4.1. Stage-wise protocol

Each run is decomposed into disjoint stages with explicit start/end timestamps. We define total distillation energy as the sum of teacher-side work, student training, and evaluation, and map these terms to logged stages:

- $E_{\text{prerun}}$ – standalone, one-time environment stabilization and smoke tests (short batch to stabilize energy and validate logging);

- $E_{\text{teacher}}$ – teacher forward passes composed of synthetic-data generation ($E_{\text{gen}}$) or logit caching ($E_{\text{logit}}$), corresponding respectively to synthetic SFT and KD;

- $E_{\text{student}}$ – student training part of the pipeline, shared across SFT, KD, and synthetic SFT;

- $E_{\text{eval}}$ – core and auxiliary evaluation suites used to construct quality metrics.

For each stage we log wall-clock time, token counts, and energy, and aggregate these into stage-wise and pipeline totals used in our results and Pareto frontiers.

### 4.2. Measurement and reporting

We treat GPU energy from NVML telemetry as the primary signal and use CodeCarbon-style estimators for CPU energy and derived $CO_2$e. We report energy in kWh and normalize by tokens processed to obtain Joules-per-token (J/token) and tokens-per-second as comparable efficiency axes. Because carbon depends on deployment-specific assumptions (e.g., PUE and regional grid intensity), we treat $CO_2$e as derived and assumption-sensitive; our main analysis focuses on directly measured energy and J/token. Full measurement details and assumptions are provided in Appendix A (Supplementary).

## 5. Experimental Design

Our experimental design was structured to answer the posed research questions under controlled and fixed hardware, software, and data conditions over the different distillation regimes and student scales to allow for reproducibility and comparable energy measurements. To facilitate reproducibility, we release the measurement harness, experiment configurations, and quality-score scripts at https://github.com/StellarLuminosity/Energy.

### 5.1. Core Grid

The central experiment was designed to cross three pipelines — baseline supervised fine-tuning, logit-based knowledge distillation (KD), and synthetic supervised fine-tuning (SFT) — with three student scales: 1B, 7B, and 13B OLMo-2 models distilled from a 32B instruction-tuned teacher. Each experimental configuration corresponds to a full pipeline

instantiated on one of the supervised workloads (instruction following, math, or code, as described in Section 3) and run end-to-end with stage-wise energy logging as defined in Section 4.

Across the core experimental conditions, we hold fixed:

- **Hardware and environment** settings, as described in Section 3.1

- **Tokenizer and preprocessing.** All models share the same OLMo-2 tokenizer and a single chat-style formatting pipeline, ensuring that the 'tokens processed' measurement is comparable across student sizes and pipelines.

- **Training configuration and stopping rule.** We use a baseline training configuration with fixed hyperparameter settings for all pipelines and student sizes to allow for reproducibility and effective comparisons. See the Appendix for more details about the exact hyperparameters and values fixed.

  Training proceeded until an early-stopping criterion is met (no improvement beyond a small tolerance of $\epsilon = 2 \times 10^{-3}$ for three consecutive evaluations); we retained the checkpoint with the best validation loss.

### 5.2. Evaluation Protocol

All trained students and the teacher are evaluated on a fixed benchmark suite that covers instruction following, dialogue, mathematical reasoning, and general knowledge: AlpacaEval 2, IFEval, MT-Bench-101, GSM8K, and MMLU. This particular suite was chosen to test general retention and broad knowledge (MMLU, GSM8K), measure domain gains and losses in math and instruction-following / dialogue (GSM8K, AlpacaEval 2, IFEval, MT-Bench-101), and align with the Tülu/OLMo training objectives and benchmark choices from the OLMo-2 paper (OLMo et al., 2024).

**Quality score.** To compare students across benchmarks of differing scales, we aggregate the five evaluation scores via equally-weighted teacher-relative retention:

$$Q_i = \frac{1}{B} \sum_{b=1}^{B} \frac{s_{i,b}}{s_{\text{teacher},b}}, \tag{3}$$

where $B = 5$, $s_{i,b}$ is the score of student model $i$ on benchmark $b$, and $s_{\text{teacher},b}$ is the corresponding score of the OLMo-2 32B teacher for benchmarks AlpacaEval 2 LC, IFEval, GSM8K, MMLU (from Ai2's published OLMo-2 evaluation table (OLMo et al., 2024)), and MT-Bench-101 (measured under our evaluation protocol) respectively. $Q_i$ measures the average fraction of teacher quality retained by the student across the evaluation suite. Raw per-benchmark scores are reported in Appendix E.

## 5.3. Ablations and Hyperparameter Sweeps

Beyond the core experiments, we run a small set of targeted ablations to probe how distillation-specific choices move models along the energy–quality frontier, keeping all other factors fixed.

**KD hyperparameters.** For logit-based KD, we vary the distillation temperature $T \in \{1.0, 2.0, 4.0\}$ and the soft-label loss weight $\alpha \in \{0.3, 0.5, 0.8\}$ in the objective of Eq. (1), primarily for the 1B and 7B students.

**Synthetic SFT generation budget.** For synthetic SFT, we vary the teacher generation budget and decoding configuration by changing: 1) the maximum number of new tokens per response, `max_new_tokens` $\in \{256, 512, 1024\}$, and 2) the number of prompts used to construct the synthetic dataset (e.g., subsampling from $\approx 7000$ to $\approx 3500$ prompts).

## 6. Results

We report end-to-end energy, quality, and Pareto frontier results for the three distillation regimes.

### 6.1. Primary Energy Frontiers

Table 2 summarizes the central experimental grid: for each pipeline and student size, we report total end-to-end energy in kWh, Joules per token (J/tok) normalized by tokens processed, and a scalar quality score $Q$ obtained by averaging normalized benchmark scores over the evaluation suite. As expected, larger students consume more energy and achieve higher quality with diminishing returns: moving from 1B to 13B roughly quintuples total energy while improving $Q$ by at most $\sim 0.15$–0.2.

*Table 2.* Full-pipeline energy breakdown; Energy values reported as means over 2–3 repeated runs; Totals exclude prerun validation.

| Pipeline | Size | E (kWh) | J/tok | $Q$ |
|----------|------|---------|-------|-----|
| Baseline SFT | 1B | 7.00 | 0.84 | 0.69 |
| | 7B | 19.50 | 2.34 | 0.90 |
| | 13B | 34.60 | 4.15 | 0.99 |
| KD distillation | 1B | 16.90 | 2.03 | 0.70 |
| | 7B | 28.40 | 3.41 | 0.78 |
| | 13B | 42.50 | 5.10 | 0.82 |
| Synthetic SFT | 1B | 16.65 | 2.00 | 0.71 |
| | 7B | 28.25 | 3.39 | 0.79 |
| | 13B | 40.70 | 4.88 | 0.85 |

Figure 1 below further summarizes end-to-end energy-quality tradeoff across the three training regimes, with the x-axis reporting full-pipeline energy per run (kWh), and the y-axis reports the normalized aggregate quality score Q

over AlpacaEval 2, IFEval, MT-Bench-101, GSM8K, and MMLU.

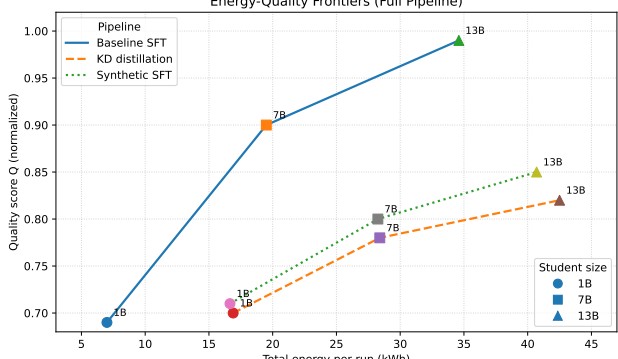

*Figure 1.* Energy-quality frontier across each pipeline's full end-to-end energy cost

As can be observed from Figure 1, the teacher side is a major cost driver in our setting: once logit caching or synthetic label generation is included, KD and synthetic SFT consume substantially more end-to-end energy without commensurate gains in aggregate $Q$. This may appear counter to the common view of distillation as a cheaper method, but that claim typically refers to student-only training cost or downstream inference efficiency; by contrast, our accounting treats teacher computation as a first-class, end-to-end cost. When teacher artifacts are generated per experiment and not reused, the teacher acts as a near-fixed overhead that shifts KD/synthetic SFT curves to the right — an effect especially pronounced for smaller models or shorter training times where the student-side compute cannot amortize the teacher work. In contrast, scaling the student under baseline SFT yields strong returns, whereas KD and synthetic SFT exhibit generally weaker scaling.

Under one-off full-pipeline accounting, baseline SFT defines the energy-dominant frontier in our measured setting. At the 1B scale, KD and synthetic SFT obtain slightly higher aggregate $Q$ than baseline SFT, but require roughly $2.4\times$ more end-to-end energy. At 7B and 13B, baseline SFT strictly dominates both distillation pipelines in the aggregate energy–quality plane.

Bigger models, such as 13B, maximize quality at moderate cost, while smaller models such as 1B minimize energy. Distillation can still be favorable in settings where teacher artifacts (cached logits or synthetic datasets) are reused across multiple students or hyperparameter sweeps, effectively amortizing the teacher-side energy and moving KD/synthetic SFT toward the Pareto frontier, particularly for instruction-following objectives. It should still be noted that since these conclusions depend on the definition of $Q$, our small model sizes, and inclusion of the full end-to-end accounting costs, it is possible that student-only energy or

per-benchmark analyses may identify regimes where distillation yields superior performance on specific capabilities even if the aggregate frontier is dominated.

Because $CO_2$e scales linearly with energy under a fixed grid/PUE factor $\gamma$ (Appendix A, Eq. 6), the frontier in Figure 1 can be read equivalently as an emissions–quality frontier. In our observations, the teacher overhead that shifts KD/synthetic SFT "to the right" in kWh produces the same shift in $CO_2$e.

### 6.2. Stage-wise Distillation Energy Breakdown

*Table 3.* Stage-wise energy breakdown (kWh)

| Stage | 1B | 7B | 13B |
|---|---|---|---|
| Prerun | 0.12 | 0.12 | 0.12 |
| **Baseline SFT (no teacher)** | | | |
| Data preprocess | 0.37 | 0.37 | 0.37 |
| Student training (SFT) | 6.30 | 18.45 | 33.15 |
| Evaluation | 0.33 | 0.68 | 1.08 |
| **KD distillation (offline)** | | | |
| Data preprocess | 0.37 | 0.37 | 0.37 |
| Logit caching | 11.00 | 11.00 | 11.00 |
| Student training (KD) | 5.20 | 16.35 | 30.05 |
| Evaluation | 0.33 | 0.68 | 1.08 |
| **Synthetic SFT** | | | |
| Data preprocess | 0.37 | 0.37 | 0.37 |
| Synthetic data generation | 10.60 | 10.60 | 10.60 |
| Student training (SFT on synthetic) | 5.35 | 16.60 | 28.65 |
| Evaluation | 0.33 | 0.68 | 1.08 |

To understand where the energy is actually spent, we decompose each run into pre-run, preprocessing, teacher-side computation, student training, and evaluation, as defined in Section 4. Table 3 shows the exact numerical breakdown of total energy kWh by stage, while the figure below visualizes the cost attribution for stages. We observe that for baseline SFT, the vast majority of energy is consumed by student training, with small contributions from preprocessing and evaluation. In contrast, KD splits its budget between teacher logit caching and student training, while synthetic SFT is dominated by teacher-side synthetic data generation; dataset preprocessing and evaluation remain relatively small in all regimes.

Stage-wise accounting makes explicit which component must be optimized to change end-to-end efficiency: for larger students, student training dominates (baseline and

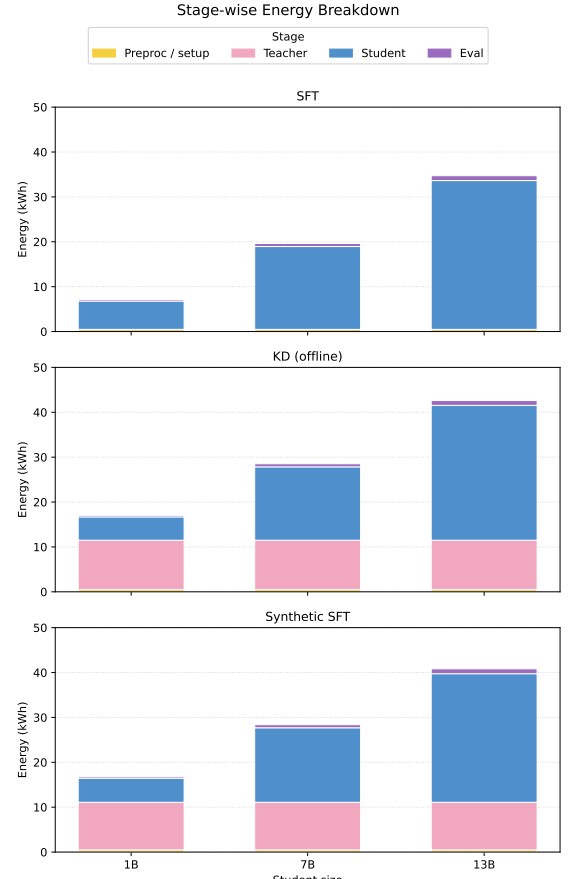

*Figure 2.* Stage-wise energy breakdown (kWh) across student sizes

teacher-mediated pipelines), while for smaller students, teacher artifact creation can be the primary driver of total energy.

We note that student-side training energy is consistently lower under KD and synthetic SFT than under baseline SFT at the same scale. This is a convergence effect, with distilled pipelines reaching the early-stopping criterion in fewer optimization steps than baseline SFT due to the additional supervision signal.

### 6.3. Teacher Reuse and Amortization

Next, we examine how the practice of reusing teacher artifacts can change end-to-end energy conclusions. Figure 3 plots the average energy per trained student as a function of the number of student runs $N$ that share a single set of cached logits (KD) or a single synthetic dataset (synthetic SFT) generated by the teacher.

Baseline SFT has no teacher-side overhead, so its per-model energy is essentially constant in $N$, dominated by student training. In contrast, KD and synthetic SFT include a near-fixed teacher artifact cost (logit caching or generation) that

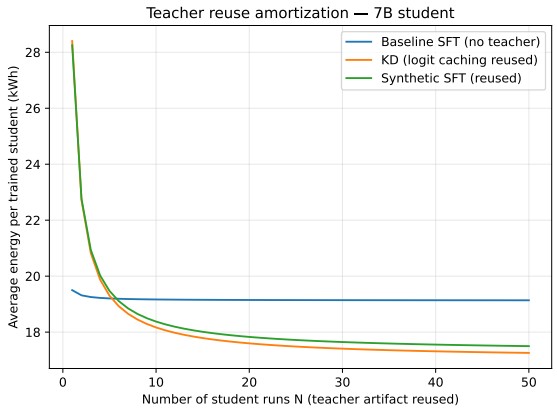

*Figure 3.* Amortizing teacher cost through reuse for 7B models

contributes as $1/N$ when averaged across runs: as $N$ grows, the amortized curves rapidly drop toward their student-only training costs.

The break-even reuse threshold admits a closed form:

$$N^* = \frac{E_{\text{teacher}}}{E_{\text{student}}^{\text{baseline}} - E_{\text{student}}^{\text{distill}}}, \qquad (4)$$

where $E_{\text{teacher}}$ is the one-time teacher artifact cost (logit caching or synthetic generation) and the denominator is the per-run training-energy gap between the baseline and the distilled student at fixed scale. Applying Eq. 4 to our measured stage-wise energies yields the following thresholds:

| Pipeline | 1B | 7B | 13B |
|---|---|---|---|
| KD | $\approx 10$ | $\approx 5$–$6$ | $\approx 4$ |
| Synthetic SFT | $\approx 11$ | $\approx 6$ | $\approx 2$–$3$ |

The pattern is systematic; smaller students require more reuse to amortize the fixed teacher cost because their per-run energy gap relative to baseline SFT is narrower, while larger students cross the break-even threshold sooner. The design rule *reuse-before-regenerate* is therefore most important at small scale, where one-off distillation is least energy-competitive. The absolute kWh values are hardware-specific, so Eq. 4 should be recomputed for new hardware, model families, or parallelism strategies.

### 6.4. Sensitivity Analysis

Figures 4, 5, and 6 summarize representative sweeps. For KD, increasing temperature $T$ provides only modest quality gains while slightly increasing energy, and some $(T, \alpha)$ settings are Pareto-dominated—wasting energy for negligible benefit. In practice, $T$ is a second-order knob: once $T$ is reasonable, $\alpha$ more directly controls the quality–energy tradeoff.

Figure 5 isolates $\alpha$ under offline KD with cached teacher

outputs. Quality increases with $\alpha$ for all sizes, while energy shifts are small because teacher-side costs are fixed and $\alpha$ mostly affects convergence. The effect is size-dependent: 1B can favor lower $\alpha$ for slightly lower energy with minimal quality loss, whereas 7B/13B benefit more from moderate-to-higher $\alpha$ with only a mild energy increase.

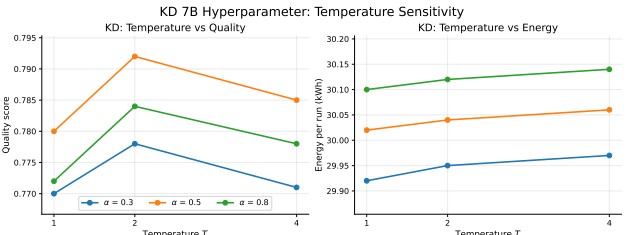

*Figure 4.* Hyperparameter sensitivity for KD 7B Student; Left: quality as a function of distillation temperature; Right: energy per run.

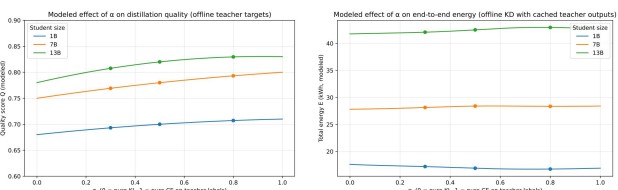

*Figure 5.* Modeled impact of the KD mixing weight $\alpha$ on quality (Q) and end-to-end energy (kWh) for 1B/7B/13B students.

For synthetic SFT, the generation budget is the dominant energy driver. Increasing `max_new_tokens` consistently improves quality, but the energy cost grows nonlinearly and exhibits diminishing returns beyond moderate lengths (Fig. 6). Reducing the number of synthetic prompts provides a direct, predictable energy reduction at comparable quality, suggesting that budget-aware synthetic distillation should prioritize moderate sequence lengths and dataset sizes before scaling generation aggressively.

### 6.5. Inference-Side Energy

Our main accounting focuses on training-time energy. For completeness, we also estimate inference energy from evaluation-stage NVML telemetry, using forward passes: 0.27 J/token for 1B, 0.68 J/token for 7B, and 1.44 J/token for 13B. The standard inference-savings argument requires a smaller distilled model to substitute for a larger baseline at comparable quality. We do not observe such cross-size equivalence as KD 1B reaches $Q = 0.70$ versus baseline SFT 7B at $Q = 0.90$, and KD 7B reaches $Q = 0.78$ versus baseline SFT 13B at $Q = 0.99$. Same-size comparisons use the same serving architecture, so the inference energy per token is the same, with the relevant difference in costs coming from the training-time pipeline.

For settings where a smaller distilled student is an acceptable

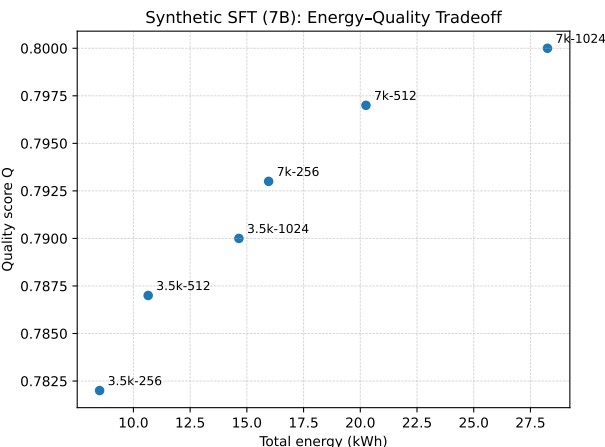

*Figure 6.* Energy–quality tradeoff for 7B synthetic distillation.

quality-equivalent substitute, the inference break-even point is

$$T^* = \frac{E_{\text{extra-train,kWh}} \cdot 3{,}600{,}000}{j_{\text{ref}} - j_{\text{student}}}, \tag{5}$$

where $E_{\text{extra-train,kWh}}$ is the additional training energy of the distilled pipeline and $j_{\text{ref}} - j_{\text{student}}$ is the per-token inference-energy saving.

## 7. Discussion and Takeaways

End-to-end accounting shows distillation is not inherently cheap. The dominant energy drivers are (i) **student compute** (model size and training tokens) and (ii) **teacher artifact creation** (logit caching or synthetic generation), which acts as a near-fixed overhead unless reused.

**1) Scale dominates.** Bigger students cost much more for diminishing gains, making this the largest lever on total kWh. Mid-scale students typically offer the better energy–quality tradeoff than defaulting to the largest.

**2) Teacher reuse can invert the ranking.** In one-off runs, teacher costs push KD and synthetic SFT right of baseline SFT on the frontier (Figure 1), especially for smaller students. With reuse across students, sweeps, or seeds, the ordering flips (Figure 3): distillation becomes energy-competitive only when teacher costs are amortized. Whether distillation is cost-efficient is therefore primarily a workflow question, not an intrinsic property of KD or synthetic SFT.

**3) Stage-wise attribution dictates the right knob.** Baseline SFT is dominated by **student training**; KD splits between **logit caching** and training; synthetic SFT is dominated by **teacher generation** (Figure 2). Effective optimization depends on the dominant stage—training tokens for SFT, generation budget for synthetic SFT, and reuse whenever a teacher stage is present.

**4) Hyperparameters are frontier controls.** For KD, some $(T, \alpha)$ pairs are Pareto-dominated. For synthetic SFT, `max_new_tokens` and dataset size yield non-linear energy growth with diminishing returns. Tune on the energy–quality frontier, not accuracy alone.

**Practical recommendations.** (i) Report both student-only and full-pipeline energy with explicit reuse assumptions. (ii) Treat teacher artifacts as shared infrastructure—cached, versioned, and published via lightweight registries logging metadata, decoding settings, licensing, and measured production energy—and adopt a default *reuse-before-regenerate* workflow. (iii) Use baseline SFT when reuse is low; favor KD or synthetic SFT when reuse and quality demands are high.

## 8. Limitations and Future Work

Our study has several limitations that qualify the scope of the empirical findings.

(1) First, all experiments were conducted on a single hardware and cluster configuration as described in 3.1. While this controlled setting was useful for isolating pipeline and hyperparameter effects, it does not capture variability across different GPU accelerators (e.g., A100, L4, TPUs), power caps, or multi-GPU training. Extending the protocol to multiple hardware devices and experimenting with power or latency-constrained regimes is a direction for future work.

(2) Our research focused on a single, fully open model family (OLMo-2) and a specific teacher–student size configurations (32B teacher, 1B/7B/13B students). Distillation behavior, including teacher-side cost, convergence speed, and quality gains, may differ for other architectures and models, pretraining corpora, and compression methods (e.g., quantization, pruning, LoRA). Likewise, different teacher size can likely shift the cost of teacher-size of distillation, the energy–quality frontier and the reuse thresholds at which distillation becomes beneficial. Applying the proposed protocol to other model families and compression strategies would test how robust these design rules are.

(3) The training and evaluation domains are limited to three pipelines on three supervised workloads (instruction following, math reasoning, and code generation), we do not cover other types such as safety alignment, multilingual tasks, long-context reasoning. Different task mixtures, deployment objectives, or evaluation protocols could lead to different trade-offs on the energy–quality frontier.

(4) Our accounting relies on specific measurement and modeling assumptions. We treat NVML-based GPU telemetry as ground truth and use estimator-based methods for CPU energy and $CO_2$e under fixed PUE and grid-intensity assumptions; deployment-specific datacenter characteristics

are not captured.

(5) Finally, the absolute kWh values, J/token values, and the 7B reuse threshold $N$ are configuration-specific. They depend on hardware, parallelism strategy, model family, decoding setup, and convergence behavior. The broader conclusion is not that these numerical thresholds transfer directly, but that teacher artifact creation is a measurable pipeline cost that can change energy–quality rankings unless it is amortized. Eq. 4 and the released energy harness are intended to make this threshold recomputable for other hardware and model families.

## 9. Conclusion

This work set out to answer a simple yet unexplored question: when is LLM distillation actually energy-efficient? While distillation and smaller students are often presented as "greener" or "cheaper" methods, existing practice rarely accounts for the full distillation pipeline, especially the teacher-side generation costs. We addressed this gap by holistically evaluating the energy demand across three pipelines, comparing baseline supervised fine-tuning, logit-based KD and synthetic SFT methods for 1B, 7B, and 13B students distilled from a 32B teacher model, and releasing a distillation energy accounting protocol and harness that allows further evaluation of distillation costs, decomposing each pipeline into stages, measuring GPU energy via NVML with complementary CPU and carbon estimates, and normalizing results in Joules per token under controlled hardware, software, and data conditions.

Our stage-wise breakdowns and energy–quality frontiers show that distillation is not inherently more sustainable: once the teacher-side costs are included, both KD and synthetic SFT often consume more end-to-end energy than a strong SFT baseline for modest quality gains. At the same time, teacher generation and logit caching costs amortize sharply with reuse; under realistic reuse levels across students or fine-tuning rounds, KD can match or beat baseline SFT, and synthetic SFT can become strictly more energy-efficient at higher quality targets.

## Impact Statement

In light of the current rapid growth in large-scale AI deployment, demand for GPUs and new datacenter capacity has risen sharply, while systematic tracking and reporting of energy is often overlooked, in spite of the substantial environmental and societal impacts of large training runs, as well as direct financial costs of operating and developing these systems. Many training and distillation decisions are still driven primarily by benchmark scores and model quality metrics rather than by a careful assessment of return on investment in compute, energy, financial spending, and emis-

sions. This disconnect risks locking in infrastructure and practices that prove theoretically efficient on benchmarks but are costly in real-world deployment settings.

This work aims to make the energy, emissions, and costs of model distillation more transparent and measurable. By providing stage-wise energy accounting, end-to-end energy–quality frontiers, and concrete design guidelines, our results can help practitioners select distillation pipelines and hyperparameters that substantially reduce operational energy use for training smaller language models. In principle, this can lower the environmental footprint of deploying capable models, encouraging more honest reporting of compute and energy in both academic and industrial settings.

At the same time, more efficient distillation pipelines may also lower the cost of training and models that are potentially harmful or misaligned. Our analysis does not address questions of content safety, misuse, or governance, and these methods could be applied to optimize the energy use of models with negative societal impacts. We therefore view our contribution as complementary to ongoing work on safety and governance: energy-aware distillation should be combined with strong safeguards on how distilled models are trained, evaluated, and deployed.

Finally, our measurements are limited to a specific hardware and software stack (H100 GPUs, OLMo-2 models, and a small set of supervised tasks) and focus on operational energy during distillation and evaluation. We do not account for embodied emissions and costs which arise from hardware manufacturing or data center infrastructure. As a result, our quantitative estimates should be interpreted as lower bounds on the full lifecycle environmental cost of large-scale language model distillation.

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

# Appendix

## A. Measurement Details

NVML provides a GPU power time series $P_{\text{GPU}}(t)$ (Watts), which we sample every 0.5s as a trade-off between noise and logging overhead. For a stage with start time $t_s$ and end time $t_e$, we approximate GPU energy by numerical integration:

$$E_{\text{GPU}} \approx \int_{t_s}^{t_e} P_{\text{GPU}}(t)\, dt.$$

CPU energy is obtained from the CodeCarbon estimator in process-tracking mode. Stage energy is computed as

$$E_{\text{total}}^{(\text{stage})} = E_{\text{GPU}} + E_{\text{CPU}},$$

and pipeline totals are computed by summing $E_{\text{total}}^{(\text{stage})}$ over stages. We report both Joules and kWh using $1\,\text{kWh} = 3.6 \times 10^6$ J.

To compare runs with different sequence lengths and token counts, we normalize energy and throughput by tokens processed. For a stage that processes $N_{\text{tokens}}$ tokens,

$$\text{J/token} = \frac{E_{\text{total}}^{(\text{stage})}}{N_{\text{tokens}}}.$$

We derive $CO_2$e using a regional grid factor $g_{\text{region}}$ (kg $CO_2$e / kWh) and a data-center PUE:

$$CO_2\text{e} \; = \; E_{\text{total,kWh}} \times \text{PUE} \times g_{\text{region}}. \qquad (6)$$

Because PUE and $g_{\text{region}}$ are deployment-dependent, we treat $CO_2$e as assumption-sensitive and emphasize direct energy measurements in the main text. Runs are repeated 2–3 times and with different seeds to estimate variance in both energy and performance.

## B. Detailed Training Configuration

Unless otherwise, we hold the following training settings fixed across *all* experiments and pipelines:

- **Optimizer**: Adafactor (memory-efficient variant of Adam; permits all experiments to run on a single GPU).

- **Learning rate**: $5 \times 10^{-5}$.

- **Scheduler**: cosine decay with 100 warmup steps.

- **Batch size**: effective batch size of 4 with 16 gradient-accumulation steps; evaluation batch size 1.

- **Gradient clipping**: maximum gradient norm 1.0.

- **Precision**: `bfloat16` for both training and evaluation.

- **Sequence length**: maximum sequence length of 1024 tokens for training and evaluation.

## C. Hardware and Software Environment

All experiments are run on the same machine type to ensure comparability of energy measurements. Each run uses a single NVIDIA H100 80GB HBM3 GPU with a power limit of 700 W, paired with an Intel(R) Xeon(R) Gold 6442Y CPU (48 physical cores; 16 used) and approximately 2 TB of system RAM. The software stack consists of Python 3.10.13, PyTorch 2.6.0+cu124, CUDA 12.4, and `transformers` 4.51.3.

## D. Total GPU Wall-Clock Time

Across the core 3×3 grid (2–3 repeats) plus the KD ($T \times \alpha$) and synthetic SFT (`max_new_tokens`) sweeps, we estimate a total of $\approx$ 2,000 **H100 GPU-hours** of wall-clock compute (about **83 GPU-days** on a single GPU), obtained by converting the summed measured kWh to time assuming an average H100 draw of $\approx$0.65 kW.

## E. Per-Benchmark Evaluation Scores

Table below reports the model per-benchmark scores used to compute the aggregate quality score $Q$ (Eq. 3, Section 5.2) for each pipeline–student-size configuration.

*Table 4.* Per-benchmark model scores

| Pipeline | Size | AE2 | IFEval | GSM8K | MMLU | MT |
|---|---|---|---|---|---|---|
| | 1B | 6 | 70 | 68 | 44 | 5.5 |
| Baseline SFT | 7B | 12 | 70 | 74 | 69 | 7.6 |
| | 13B | 17 | 75 | 78 | 70 | 7.8 |
| | 1B | 6 | 69 | 67 | 43 | 6.2 |
| KD | 7B | 10 | 62 | 76 | 55 | 6.0 |
| | 13B | 10 | 70 | 74 | 58 | 6.6 |
| | 1B | 6 | 69 | 68 | 43 | 6.2 |
| Synthetic SFT | 7B | 10 | 61 | 77 | 55 | 6.5 |
| | 13B | 11 | 70 | 75 | 60 | 6.8 |

The per-benchmark scores show that distillation can provide modest task-specific gains; however, these gains do not reverse the aggregate energy–quality conclusions stated in the main text. Practitioners targeting a specific capability should therefore recompute the frontier using task-appropriate quality metrics; the accounting framework itself is independent of the choice of metric.

