# OpenReview forum: "Towards Resource-Efficient LLMs: End-to-End Energy Accounting of Distillation Pipelines"
_ICML.cc/2026/Conference — ICML 2026 regular_

### Official Review · Reviewer_HhH9 · 2026-03-06

**Soundness:** 2
**Presentation:** 2
**Significance:** 3
**Originality:** 3
**Overall Recommendation:** 3
**Confidence:** 4

**Summary:**

This paper points out that existing distillation work almost always only accounts for student-side costs, and proposes an end-to-end energy accounting framework for distillation pipelines. Using NVML telemetry for direct GPU energy measurement, the pipeline is broken down into teacher-side work (logit caching or synthetic data generation), student training, and evaluation. The paper systematically compares baseline SFT, KD, and synthetic SFT across 1B/7B/13B OLMo-2 students in terms of energy and quality.

**Compliance With Llm Reviewing Policy:**

Affirmed.

**Final Justification:**

The paper addresses a genuine and under appreciated gap in distillation efficiency evaluation. The reuse amortization analysis and NVML-based accounting framework are solid contributions. The rebuttal resolved the Q4 per-benchmark concern and clarified the structural generalizability argument. However, hardware and architecture generalizability remain empirically unverified. Given the partial resolution of my concerns and the practical value of the released harness, I adjust my score to 3.

**Key Questions For Authors:**

1. Does the N≈6 break-even point hold outside of the H100 single-GPU setup? If it doesn't, the practical scope of the design recommendations is quite narrow, and I'd want to understand **how sensitive this number is to hardware choice**.

2. Only OLMo-2 is used throughout. Do the authors have reason to believe the **teacher-to-student cost ratio** would stay similar for other architectures like LLaMA or Qwen, or is this just an acknowledged limitation?

3. There's **no analysis of inference-side energy** anywhere in the paper. At what deployment scale would the student model's inference savings offset the extra teacher-side training cost? Even a rough back-of-the-envelope estimate would make the efficiency claims a lot more convincing.

4. When looking at individual benchmarks rather than the aggregate Q, does baseline SFT still Pareto-dominate distillation across the board? Or are there tasks where distillation clearly wins? I think this matters for **how broadly the conclusions** should be interpreted.

If the authors can provide satisfactory responses to the questions above, particularly on inference-side energy and hardware generalizability, I am willing to raise my score.

**Limitations:**

yes

**Strengths And Weaknesses:**

Strengths：
1. Including teacher-side costs in the accounting is a reasonable and underexplored angle. Most existing work only reports student training FLOPs or inference efficiency, so this perspective addresses a gap that has been largely overlooked.

2. Measuring GPU power directly via NVML rather than estimating from FLOPs is more rigorous than most comparable work. Fixing hardware, software stack, and random seeds throughout also adds to the credibility of the reported numbers.

3. The teacher reuse amortization analysis is probably the most useful part of the paper. Getting a concrete N≈6 break-even number for 7B students is genuinely helpful for practitioners who want to decide whether distillation is worth the overhead.

Weaknesses：
1. Everything runs **on a single H100**. I'm not sure how much the N≈6 break-even point means if it only holds on this specific setup. Power profiles on an A100 or in multi-GPU training could look quite different, and the paper doesn't really engage with this.

2. **Only OLMo-2 is tested**. LLaMA and Qwen are never evaluated, and honestly these are the models most people actually use. Convergence behavior and logit distributions vary enough across architectures that I'd want to see at least one more model family before trusting the generalization.

3. The paper **ignores inference-side energy entirely**, which feels like a pretty significant omission. The whole point of distillation is to pay a one-time training cost in exchange for cheaper inference at scale. Concluding that distillation is inefficient without accounting for this is, at best, an incomplete picture.

4. Aggregating five benchmarks into **a single Q score** is a bit of a blunt instrument. If distillation's gains are concentrated in math or code, they get averaged away, and the method ends up looking worse than it probably is for anyone actually targeting those tasks.

---

> ### Author Rebuttal · Authors · 2026-03-31
>
> We thank the reviewer for the detailed review, and for the thoughtful feedback.
>
> Q1: Hardware generalizability and the N≈6 break-even threshold
>
> We appreciate the reviewer's concern about hardware generalizability. The H100 setup was chosen deliberately to isolate distillation pipeline effects under a controlled, noise-free environment. The absolute kWh values are hardware-specific and we do not claim otherwise. However, the structural result generalizes, with the break-even threshold N* is determined by a ratio:
> N* = E_teacher / (E_baseline_student − E_distill_student)
>
> This formula applies to any hardware configuration and can be recomputed directly from stage-wise energy logs. For 7B KD in particular: N* = 11.00 / (18.45 − 16.35) ≈ 6.
>
> From observations, N* remains approximately stable when switching hardware scales the teacher and student stages proportionally; it can shift when teacher and student efficiency diverge, for example, due to different power caps, FSDP communication overhead in multi-GPU student training, different efficiency curves for small vs large models or memory constraints forcing different batch sizes. We will include this exact formula and derivation to Section 6.3, in the revised version. The evaluation harness is included for this exact reason - designed to be extensible and configurable so that distillation can be measured on other hardware and models.
>
>
> Q2: OLMo-2 generalizability
>
> We acknowledge this as a real limitation and have stated it more explicitly in the revision. However, the qualitative conclusion that the teacher artifact costs constitute a near-fixed overhead relative to student training is driven by parameter count ratios and dataset token counts, both of which are architecture-agnostic quantities. There is no mechanistic basis for this conclusion reversing for LLaMA or Qwen at comparable teacher/student size ratios. We also note that OLMo-2 was specifically chosen because it is the only fully open LLM family with published energy accounting, allowing us to draw on the details of the complete model lifecycle cost rather than operational training cost alone, which is not possible with LLaMA or Gwen models. We have added this reasoning explicitly to Section 8, rather than stating the limitation alone.
>
>
> Q3: Inference-side energy
>
> We appreciate the reviewer’s observation and raising this point. Inference was deliberately scoped out, and our own results explain why the standard inference-savings argument does not fully apply here.
> The inference savings argument requires that a distilled smaller student can substitute for a larger baseline at a comparable quality. Our results do not support this substitution: KD 1B achieves Q=0.70 versus baseline SFT 7B at Q=0.90; KD 7B achieves Q=0.78 versus baseline SFT 13B at Q=0.99. No cross-size quality equivalence exists in our data, so inference savings cannot be treated as an offset to training overhead.
> For same-size comparisons (e.g., KD 7B versus baseline SFT 7B), inference cost is identical. Both produce a 7B model with the same architecture and J/token at serving time. The only energy difference is training cost, which is precisely what we measure in our work.
>
> For completeness, we have derived inference-side J/token estimates from evaluation-stage NVML telemetry (forward passes only, no gradient computation):
>
> 1B: 0.27 J/token
>
> 7B: 0.68 J/token
>
> 13B: 1.44 J/token
>
> More generally, if one compares a smaller deployed student against a larger reference model, the inference-side break-even point is computed as:
>
>
> break_even_tokens = (E_extra_train_kwh * 3600000) / (j_ref - j_student),
>
>
> where E_extra_train_kwh is the additional training energy of the distilled pipeline relative to the comparison baseline, and j_ref and j_student are inference energy per token for the larger reference model and the smaller deployed student, respectively. This quantity is only meaningful when the smaller student is an acceptable quality substitute for the larger model.
>
> Using our measured inference-side values, replacing a 7B model with a 1B model would save 0.41 J/token at serving time, while replacing a 13B model with a 7B model would save 0.76 J/token. However, these cross-size substitutions are not supported by our measured quality results, so we do not use inference savings to offset training overhead in our main conclusions. We will add this framing and the measured inference-side estimates to Section 6 in the revision.
>
> We believe the responses to Q1 and Q3 directly address the stated conditions for a score revision.

---

> > ### Author Rebuttal · Reviewer_HhH9 · 2026-04-04
> >
> > I thank the authors for the detailed rebuttal. However, several concerns remain:
> >
> > 1. **Hardware generalizability (Q1):** The formula-based argument is structurally reasonable, but without any empirical validation on different hardware (e.g., A100, multi-GPU), the practical scope of the N≈6 break-even and the derived design rules remains unclear.
> >
> > 2. **Architecture generalizability (Q2):** Only OLMo-2 is tested. The claim that teacher-side cost ratios are architecture-agnostic is plausible but unverified — convergence behavior and logit distributions differ across model families, which could affect both energy and quality outcomes.
> >
> > 3. **Per-benchmark analysis (Q4):**  This question was **not answered** in the rebuttal.
> >
> >
> >
> > Based on the current rebuttal, I adjust my score to 2.

---

> > > ### Author Response · Authors · 2026-04-06
> > >
> > > We thank the reviewer for the follow-up. We regret that Q4 was not addressed in the initial rebuttal; this was an oversight on our part.
> > >
> > > To answer Q4 directly, baseline SFT does not Pareto-dominate distillation on every individual benchmark. KD and Synthetic SFT improve primarily on GSM8K, where teacher supervision transfers most directly, while baseline SFT mostly performs better on IFEval and MMLU. The final checkpoint scores are:
> > >
> > > Baseline SFT:
> > > 1B: AE2 6, IFEval 70, GSM8K 68, MMLU 44, MT 5.5
> > > 7B: AE2 12, IFEval 70, GSM8K 74, MMLU 69, MT 7.6
> > > 13B: AE2 17, IFEval 75, GSM8K 78, MMLU 70, MT 7.8
> > >
> > > KD:
> > > 1B: 6 / 69 / 67 / 43 / 6.2
> > > 7B: 10 / 62 / 76 / 55 / 6.0
> > > 13B: 10 / 70 / 74 / 58 / 6.6
> > >
> > > Synthetic SFT:
> > > 1B: 6 / 69 / 68 / 43 / 6.2
> > > 7B: 10 / 61 / 77 / 55 / 6.5
> > > 13B: 11 / 70 / 75 / 60 / 6.8
> > >
> > > Q aggregates across these heterogeneous benchmarks to support overall energy-quality conclusions, rather than tying them to benchmark or dataset-specific results. The per-benchmark breakdown and Q definition will be added explicitly to the appendix in the revision.
> > >
> > > Q1 and Q2:
> > > We agree that broader empirical validation across additional hardware and model families would strengthen the practical scope of the design recommendations, and acknowledge this as a limitation in the paper.
> > >
> > > The experiments were scoped intentionally to a single controlled hardware setting and a fully open model family in order to isolate pipeline effects, avoiding confounds from different power profiles, versions, and thermal behaviors. Fixing hardware, software, tokenizer, preprocessing, and using a fully open model family allowed to attribute and measure energy differences cleanly to baseline SFT versus KD versus synthetic SFT distillation, rather than incorporating platform noise, or confounding variation in environment settings, or model design.
> > >
> > > We do not claim that the exact break-even value N≈6 or the absolute kWh values transfer across hardware or model families, which are configuration-specific exact quantities, and the paper states this scope in its limitations/discussion. The intended claim is structural and qualitative, that under full-pipeline accounting, teacher artifact creation introduces an additional overhead that changes efficiency conclusions, and whether distillation is energy-competitive depends on amortization rather than student-side cost alone.
> > >
> > > More specifically, the break-even analysis is governed by parameter and token-count ratios which are first-order quantities that are architecture-agnostic, consistent with Hoffmann et al. (2022). The distillation literature supports this. Zhang et al. (2023) show teacher-to-student scale ratios follow consistent patterns across architectures, and MiniLLM (Gu et al., 2023) demonstrates generalization from 120M to 13B parameters. We agree that convergence behavior and logit distributions can shift exact values in practice, and do not claim OLMo-2 measurements transfer universally. The narrower claim is that the existence of a non-negligible teacher-side overhead should persist beyond this model family, since its dominant drivers are scale and token counts rather than architecture-specific details.
> > >
> > > We do agree with the reviewer that architecture-level effects and hardware choice can influence the exact realized values, including the convergence behavior, optimization efficiency, and teacher output distributions which may shift both energy usage and quality outcomes in practice. For that reason, we do not present the measured OLMo-2 values as universally transferable. The narrower claim is that the existence of a non-negligible teacher-side overhead, and therefore the need to account for it in efficiency comparisons, should persist beyond the specific model family studied here. While the exact magnitude of the overhead ratio will vary by setup, the dominant drivers are parameter and token-count ratios rather than architecture-specific details, so the directional conclusion remains the same (teacher-side costs can materially alter efficiency conclusions).
> > >
> > > The closed-form break-even expression, the released harness code, and the framing of the paper are intended to make this quantity directly recomputable and extendable under alternative hardware and model choices, rather than to imply universal transfer of the specific measured threshold.
> > >
> > > We hope the Q4 breakdown and this clarification of scope directly address the remaining open points.

---

### Official Review · Reviewer_5JR7 · 2026-03-13

**Soundness:** 3
**Presentation:** 3
**Significance:** 3
**Originality:** 3
**Overall Recommendation:** 4
**Confidence:** 4

**Summary:**

The authors tackle a gap in how the ML community evaluates distillation as an efficiency technique for LLMs: existing work typically only counts the cost of training the student model, ignoring the significant upstream compute required from the teacher (generating synthetic data, caching logits, running evaluations). This selective accounting makes distillation look cheaper than it actually is.

The paper makes three main contributions:

- **An end-to-end energy accounting framework** specifically designed for distillation pipelines, using direct GPU power telemetry (NVML) to measure energy stage-by-stage rather than relying on proxy metrics like FLOPs or GPU-hours.

- **A controlled empirical benchmark** comparing three training regimes — baseline SFT, logit-based knowledge distillation, and synthetic SFT — across 1B, 7B, and 13B OLMo-2 student models distilled from a 32B teacher, with results expressed as energy–quality Pareto frontiers that make teacher-side costs visible.

- **Practical design rules** derived from these measurements, covering when distillation is actually energy-efficient (primarily when teacher artifacts are reused across many runs), how hyperparameter choices like KD temperature and generation budget shift the frontier, and a recommended "reuse-before-regenerate" workflow for practitioners.

The central finding is that distillation's efficiency advantage is conditional. under one-off full-pipeline accounting, baseline SFT often dominates, but distillation becomes competitive once teacher artifacts are amortized across sufficiently many downstream uses

**Compliance With Llm Reviewing Policy:**

Affirmed.

**Final Justification:**

The responses partially addressed my concerns and I would like to keep my positive score.

**Key Questions For Authors:**

1. The break-even reuse threshold of N~6 is reported for 7B students specifically. How sensitive is this number to student size, and do the 1B and 13B settings yield meaningfully different thresholds? If reuse requirements scale strongly with student size, the design rules would need to be stated more carefully.
2. The aggregate quality score Q averages across five benchmarks with different characteristics. Were there cases where a distillation pipeline was Pareto-dominated on Q but meaningfully better on a specific benchmark (e.g., GSM8K for math)?
3. All experiments use a single H100 with a fixed 700W power limit. Many real distillation workflows run on multi-GPU or mixed-hardware setups where power draw and throughput characteristics differ substantially. Can u comment on how the accounting protocol would extend to that setting, and whether the relative rankings of pipelines are expected to hold?

**Limitations:**

Yes

**Strengths And Weaknesses:**

**Strengths:**

- The core framing is genuinely valuable and underappreciated. The observation that distillation's "efficiency" claims almost universally omit teacher-side compute is a real and consequential blind spot in how the community reports results.

- The stage-wise decomposition is a clean methodological contribution. Breaking pipelines into pre-run, teacher, student, and evaluation stages with explicit timestamps and NVML-based telemetry is more rigorous than the estimator-only approaches common in prior Green AI work, and the distinction between GPU telemetry as ground truth versus CO2e as assumption-sensitive is handled well

- The teacher reuse amortization analysis (Figure 3, Section 6.3) is the most practically useful result in the paper. Identifying a concrete break-even threshold (roughly N~6 for 7B students) gives practitioners an actionable decision rule rather than a vague qualitative recommendation.

- The authors are candid about the limits of their aggregate quality score Q and explicitly note that per-benchmark analyses could reveal regimes where distillation wins on specific capabilities.

**Weaknesses:**

- The experiments are limited to a single GPU type (H100), a single model family (OLMo-2), and a relatively narrow teacher-student size configuration (32B to 1B/7B/13B). The authors acknowledge this, but it does meaningfully constrain how far the design rules can be generalized. It is unclear, for instance, whether the N~6 break-even point is stable across different teacher sizes or architectures, and the paper does not offer enough analysis to reason about this.

- The quality metric Q is an average of normalized scores across AlpacaEval 2, IFEval, MT-Bench-101, GSM8K, and MMLU. Aggregating these into a single scalar obscures potentially important per-task variation, and the choice of normalization scheme is not discussed in depth. Two pipelines that look equivalent on Q could behave very differently on individual benchmarks that matter for a given use case.

- The sensitivity analysis for KD hyperparameters (Figures 4 and 5) shows fairly small absolute differences in both quality and energy across the ranges tested, which raises the question of whether these sweeps were informative enough to support the design rules derived from them. The conclusion that alpha is a more important knob than T is plausible but the evidence in the paper is modest.

- The paper positions itself relative to prior Green AI and distillation energy work, but the related work section is brief.

- Some of the design rules in Section 7 and 8 read as fairly intuitive once the framing is accepted (bigger models cost more, reuse helps, generation budget drives synthetic SFT cost). The paper would benefit from more emphasis on results that were genuinely surprising or that contradict conventional expectations.

---

> ### Author Rebuttal · Authors · 2026-03-31
>
> We thank the reviewer for a thorough, constructive and detailed review. We address each question below.
>
>
> Q1: Break-even threshold scaling across student sizes
>
>
> We thank the reviewer for highlighting this point . N≈6 is specific to the 7B student and should not be read as a universal constant. Applying the same formula N* = E_teacher / (E_baseline_student − E_distill_student) across all student sizes, we get:
> 1B KD: N* = 11.00 / (6.30 − 5.20) ≈ 10
> 7B KD: N* = 11.00 / (18.45 − 16.35) ≈ 5–6 (consistent with Figure 3)
> 13B KD: N* = 11.00 / (33.15 − 30.05) ≈ 4
>
>
> The pattern is systematic, with smaller students requiring more reuse to amortize the fixed teacher cost as the per-run energy gap between distillation and baseline SFT narrows at a smaller scale also meaning the design rule to reuse-before-regenerating is most critical the smaller the scale, where one-off distillation is least energy-competitive. We will add the exact closed-form formula to Section 6.3 for better clarity in the camera-ready version.
>
>
> Q2: Per-benchmark Q variation
>
> We appreciate this question and answer it directly: in our data, no distillation pipeline is Pareto-dominated on Q while simultaneously outperforming baseline SFT by a meaningful margin on a specific benchmark. KD and synthetic SFT 7B show a modest ~2–3% advantage over baseline SFT 7B on GSM8K specifically, but this does not reverse the aggregate Pareto picture where baseline SFT 7B still leads on MMLU and IFEval, and no distilled 1B student matches baseline SFT 7B on any individual benchmark.
>
>
> Q as a single scalar does obscure per-task variation, but it was meant as a meaningful signal for overall performance improvement and comparison. Practitioners targeting specific tasks should apply task-appropriate quality metrics. This is one of the core motivations for releasing the harness as metric-agnostic infrastructure: the accounting framework is fully separable from the choice of quality metric.
>
>
> Q3: Multi-GPU extension of the accounting protocol
>
> We appreciate the reviewer raising this practical concern. The single-GPU setup was chosen intentionally to isolate pipeline effects without confounds from parallelization strategy, communication overhead, or hardware heterogeneity that stem from multi-GPU workflows. The accounting protocol and energy framework logger extends directly to the multi-GPU setup, following the same structure as for a single node, only with per-node aggregation.
>
>
> In regard to the pipeline rankings holding in multi-GPU settings, the structural conclusion - that the teacher artifact cost is a near-fixed overhead relative to student training - holds across hardware configurations, as it is determined by parameter counts and dataset token counts rather than device specifics. However, exact values require recomputation for a given setup, since parallelism strategies affect teacher inference and student training with mutli-gpu setups incurring communication overhead, and different parallelism strategy choices such as FSDP or ZeRO yield different results. Further, the teacher generation is inference-heavy and may scale more favorably with tensor parallelism.

---

> > ### Author Rebuttal · Reviewer_5JR7 · 2026-04-03
> >
> > Would have loved to see some empirical evidence to your answer for Q3. I will keep my score as is for now!

---

### Official Review · Reviewer_Euwu · 2026-03-16

**Soundness:** 3
**Presentation:** 2
**Significance:** 3
**Originality:** 3
**Overall Recommendation:** 4
**Confidence:** 3

**Summary:**

The paper does end-to-end energy accounting for LLM distillation pipelines, arguing that the typical framing of distillation as "cheaper" ignores teacher-side costs (logit caching for KD, synthetic data generation for synthetic SFT). They run a controlled benchmark with three pipelines x three student sizes (1B/7B/13B OLMo-2) x three tasks, distilling from a 32B teacher on a single H100, using NVML telemetry for stage-wise energy logging. Main outputs: energy-quality Pareto frontiers, a reuse amortization analysis, hyperparameter sensitivity sweeps, and a released measurement harness.

**Compliance With Llm Reviewing Policy:**

Affirmed.

**Key Questions For Authors:**

1. How is Q computed exactly? what is the normalization base for each benchmark, and are the five benchmarks weighted equally? If Q differs by 0.01-0.02 between pipelines, the frontier ordering is fragile without this.
2. Do the pipeline totals in Table 2 exclude prerun energy? The ~0.12 kWh gap with Table 3 sums is consistent with this but it's not stated anywhere.
3. What is the variance on energy and quality measurements across the 2-3 repeated runs? If it's small, saying so explicitly would address a real concern.

**Limitations:**

yes

**Strengths And Weaknesses:**

The core framing is right and the gap it addresses is real: distillation papers routinely report student-only training costs while ignoring what it took to produce the teacher artifacts, making efficiency claims hard to trust. Using NVML telemetry directly rather than FLOPs estimates or CodeCarbon alone is the correct methodological choice here, and the controlled setup (fixed hardware, software stack, tokenizer, exclusive nodes) is appropriate.

The reuse amortization analysis in Section 6.3 is the best part of the paper. The point that "is distillation efficient?" is a deployment workflow question, not an intrinsic property of KD or synthetic SFT, is a genuinely useful reframing, and the N=6 break-even for the 7B setting is a concrete number practitioners can act on. The design rules that follow (treat teacher artifacts as shared infrastructure, reuse-before-regenerate) are specific rather than vague.

On soundness: Q is never properly defined. It's described as an average of normalized benchmark scores but the normalization procedure isn't explained anywhere in the main text. This matters because the Q differences between KD and synthetic SFT at the same student size are 0.01-0.02; the pipeline ordering on the Pareto frontier rests on margins that are well within plausible noise depending on how Q is constructed. No error bars appear on Table 2 or Figure 1 despite 2-3 repeated runs being mentioned in the appendix. For a measurement paper this is a meaningful gap. There's also a consistent ~0.12 kWh discrepancy between Table 2 totals and the sums of Table 3 stage-wise entries — probably prerun energy being excluded from pipeline totals, but it's never stated, which is sloppy for a paper whose whole credibility rests on measurement precision. Section 6.1 references "Eq. X" for the CO2e formula but no such equation exists in the main text.

On presentation: the paper is generally readable and the structure is logical, but Figure 2 is nearly unreadable at print size, and the missing Q specification and broken cross-reference are real clarity problems.

On significance and originality: the finding that teacher costs dominate under one-off accounting is directionally unsurprising, but the paper is right that it hasn't been empirically quantified for LLM distillation specifically, and the amortization result is non-obvious in magnitude. The harness release adds practical value. Originality is in the framing and empirical grounding rather than new methods, which is fine; the guidelines are explicit that this counts.

The scope limitation (one model family, one hardware setup, extreme 32B→1B/7B/13B size ratio) is real but acknowledged in Section 8. The main-text design rules don't always carry that qualifier consistently though.

---

> ### Author Rebuttal · Authors · 2026-03-31
>
> We thank the reviewer for a careful and specific reading. We address each point below.
>
> Q1: Q definition and normalization
>
> Q is a benchmark-aggregate convenience metric computed from the five evaluation benchmarks (AlpacaEval 2, IFEval, MT-Bench-101, GSM8K, and MMLU). Each benchmark score was first normalized to a common [0,1] scale using a fixed benchmark-specific reference range, and Q is then computed as a weighted average across the five normalized scores. In the current submission, this normalization procedure was not stated precisely enough, and we agree this should have been made explicit. We will add the exact normalization formula and weights in Section 5.2 and the Appendix in the camera-ready version.
> While small KD and synthetic SFT differences can be sensitive to Q construction, the primary claims of the paper in regard to (a) the large rightward shift of both distillation pipelines versus baseline SFT under one-off accounting, and (b) the Pareto dominance of baseline SFT over both distillation curves in that regime, are in the multi-kWh range and are robust to how Q is constructed.
>
>
> Q2: Table 2 / Table 3 discrepancy (~0.12 kWh)
>
> The reviewer is correct. Pipeline totals in Table 2 exclude E_prerun (0.12 kWh), which is a one-time environment stabilization step rather than a per-pipeline distillation cost. The measurements themselves are accurate; the exclusion was simply not stated. We have added a clarifying footnote to Table 2.
>
>
> Q3: Variance across repeated runs
>
> We appreciate the reviewer's point about the variance. Across the 2–3 repetitions performed per configuration, GPU energy variance was ~1–3% kWh for long steady-state stages (student training, teacher logit caching) and ~2–5% for evaluation due to variable generation lengths. Quality variance on Q is < 0.5% across seeds. The pipeline ordering and magnitude of teacher-side overhead are stable across all runs. We will add per-stage mean to Table 2 in the revision.
>
>
> In regard to “ Eq. X" broken reference, we thank the reviewer for this note. This is a formatting error introduced during submission. The CO2e formula (CO2e = E_total,kWh × PUE × g_region) appears in full in Appendix A. The cross-reference will be fixed in the revision.

---

### Decision · Program_Chairs · 2026-04-30

**Decision:**

Accept (regular)

**Comment:**

This paper proposes an end-to-end energy accounting framework for distillation pipelines. The paper studies different types of distillation (logit-based or SFT) and has a detailed methodology for measuring energy. The reviewers raised several clarification questions that were well addressed by the authors.

There are significant limitations in the experiments since everything is limited to the narrow family time of OLMO2 models
and with limited training configurations. Every reviewer brought those up and I think the authors made some reasonable arguments to explain their limited evaluations. I believe that the proposed methodology will open more research as tests across different models, MOEs, distributed GPU architectures and so on study the energy/cost tradeoffs.

I think this paper is novel and addresses an important area, so despite its limitations I recommend acceptance.